# *Staphylococcus aureus* Infection-Related Glomerulonephritis with Dominant IgA Deposition

**DOI:** 10.3390/ijms23137482

**Published:** 2022-07-05

**Authors:** Mamiko Takayasu, Kouichi Hirayama, Homare Shimohata, Masaki Kobayashi, Akio Koyama

**Affiliations:** 1Department of Nephrology, Tokyo Medical University Ibaraki Medical Center, Ami 300-0395, Ibaraki, Japan; t-mamiko@tokyo-med.ac.jp (M.T.); h-shimo@tokyo-med.ac.jp (H.S.); masaki-k@tokyo-med.ac.jp (M.K.); 2Emeritus Professor, University of Tsukuba, Tsukuba 305-8577, Ibaraki, Japan; a-koyama@cap.ocn.ne.jp

**Keywords:** *Staphylococcus aureus*, rapidly progressive glomerulonephritis, IgA-dominant glomerulonephritis, Staphylococcus infection-associated glomerulonephritis, bacterial superantigen, T-cell receptor, cytokine, polyclonal activation

## Abstract

Since 1995, when we reported the case of a patient with glomerulonephritis with IgA deposition that occurred after a methicillin-resistant *Staphylococcus aureus* (MRSA) infection, many reports of MRSA infection-associated glomerulonephritis have accumulated. This disease is being systematized as Staphylococcus infection-associated glomerulonephritis (SAGN) in light of the apparent cause of infection, and as immunoglobulin A-dominant deposition infection-related glomerulonephritis (IgA-IRGN) in light of its histopathology. This glomerulonephritis usually presents as rapidly progressive glomerulonephritis or acute kidney injury with various degrees of proteinuria and microscopic hematuria along with an ongoing infection. Its renal pathology has shown several types of mesangial and/or endocapillary proliferative glomerulonephritis with various degrees of crescent formation and tubulointerstitial nephritis. IgA, IgG, and C_3_ staining in the mesangium and along the glomerular capillary walls have been observed on immunofluorescence examinations. A marked activation of T cells, an increase in specific variable regions of the T-cell receptor β-chain-positive cells, hypercytokinemia, and increased polyclonal immune complexes have also been observed in this glomerulonephritis. In the development of this disease, staphylococcal enterotoxin may be involved as a superantigen, but further investigations are needed to clarify the mechanisms underlying this disease. Here, we review 336 cases of IgA-IRGN and 218 cases of SAGN.

## 1. Introduction

Staphylococci have been identified as causal agents in the genesis of glomerulonephritis. Most reports linking Staphylococci in infection-related glomerulonephritis (IRGN) have emphasized two clinical forms: *Staphylococcus epidermidis* (*S. epidermidis*) bacteremia with an infected ventriculojugular shunt [1], and *Staphylococcus aureus* (*S. aureus*) bacteremia with endocarditis [2]. These types of glomerulonephritis are caused by the deposition of immune complexes composed of immunoglobulin (Ig)G antibodies and bacterial antigens in the glomeruli. In contrast to those types of glomerulonephritis, in 1995 we reported the case of a patient with glomerulonephritis which had IgA deposition that occurred after a methicillin-resistant *S. aureus* (MRSA) infection [3]. Since then, many reports of MRSA infection-associated glomerulonephritis have accumulated. Most of the initial reports of MRSA infection-associated glomerulonephritis were from Japan; more recently, however, cases in the U.S. and in Europe have been described [4].

The same clinical features as MRSA infection-associated glomerulonephritis were observed in some cases with a methicillin-sensitive *S. aureus* or *S. epidermidis* infection. These types of glomerulonephritis are thus called Staphylococcus infection-associated glomerulonephritis (SAGN) [5,6]. The most common pattern observed in SAGN has consisted of mild-to-moderate IgA and moderate-to-strong complement factor 3 (C_3_) staining on immunofluorescence observation, with weak or no IgA in 25% of the cases. SAGN is thus not always IgA-dominant [6].

In all of the cases of MRSA infection-associated glomerulonephritis identified by our research group, the deposition of IgA has been remarkable; this feature differs from those of typical postinfection nephritis, which is characterized by glomerular deposition of either C_3_ and IgG, or of C_3_ only. Since our 1995 report [3], many descriptions of the same histological features as this glomerulonephritis have accumulated, and the features have been observed in cases with and without staphylococcal infection [7]. These types of glomerulonephritis are thus called IgA-dominant deposition infection-related glomerulonephritis (IgA-IRGN) [7,8].

In this review, we delineate the characteristics of SAGN and IgA-IRGN, and we describe the pathogenesis of SAGN with IgA-dominant deposition.

## 2. Review Methods

### 2.1. Literature Search

We searched the available literature in the following electronic databases: PubMed/MEDLINE, EMBASE, and Web of Science: Science Citation Index Expanded. The key words that were used included “Staphylococcus” and “infection” or “IgA-dominant” and “glomerulonephritis”.

Two hundred fifty-four reports were detected, and we collected the 206 reports that were published between our 1995 report [3] and 31 December 2021. Among the 206 reports, 20 reports that were in a language other than English, as well as a single study on birds (a hyacinth macaw), were excluded. Twenty-one other reports were excluded because of the publication type (e.g., review). Among the remaining 164 reports, 39 were excluded because the subjects had different diseases: nonrenal diseases (*n* = 13), hyper-IgE syndrome (*n* = 1), end-stage kidney disease (*n* = 6), urinary tract infection without glomerulonephritis (*n* = 2), AA amyloidosis (*n* = 1), lupus nephritis (*n* = 2), antineutrophil cytoplasmic antibody (ANCA)-associated vasculitis (*n* = 12), and antiglomerular basement disease (*n* = 2). One report of drug (rifampicin)-induced glomerulonephritis and five reports of noninfectious kidney diseases were excluded. Five nonclinical studies and two studies with a pool analysis of the literature [5,7] were excluded. Five reports of Staphylococcus aureus infection-related glomerulonephritis with IgA-dominant deposition [9,10,11,12,13] among our 12 reports of infection-related glomerulonephritis will be presented here together. Two pre-2000 cohort studies that were neither IgA-IRGN nor SAGN were also excluded. A final total consisting of 62 case reports (Appendix A) [14,15,16,17,18,19,20,21,22,23,24,25,26,27,28,29,30,31,32,33,34,35,36,37,38,39,40,41,42,43,44,45,46,47,48,49,50,51,52,53,54,55,56,57,58,59,60,61,62,63,64,65,66,67,68,69,70,71,72,73,74,75], 36 cohort studies [76,77,78,79,80,81,82,83,84,85,86,87,88,89,90,91,92,93,94,95,96,97,98,99,100,101,102,103,104,105,106,107,108,109,110,111], and our combined reports (Appendix A) of IgA-IRGN or SAGN was obtained and qualified.

### 2.2. Categorize of Selected Cases

In the 36 cohort studies [76,77,78,79,80,81,82,83,84,85,86,87,88,89,90,91,92,93,94,95,96,97,98,99,100,101,102,103,104,105,106,107,108,109,110,111], the subjects had been identified with the use of varying definitions (Appendix A). We categorized these studies as follows: IgA-IRGN or not, and SAGN or not. We analyzed the 62 case reports [14,15,16,17,18,19,20,21,22,23,24,25,26,27,28,29,30,31,32,33,34,35,36,37,38,39,40,41,42,43,44,45,46,47,48,49,50,51,52,53,54,55,56,57,58,59,60,61,62,63,64,65,66,67,68,69,70,71,72,73,74,75] of patients with Staphylococcus infection-associated glomerulonephritis with dominant IgA deposition as both IgA-IRGN and SAGN, and we also categorized these reports as IgA-IRGN or not and SAGN or not. We integrated and analyzed our own papers [3,9,10,11,12,13] as both IgA-IRGN and SAGN. Thus, 336 cases of IgA-IRGN and 218 cases of SAGN were analyzed.

## 3. Clinical Features

### 3.1. Epidemiology and Characteristics

The incidences of both IgA-IRGN and SAGN are difficult to determine, as there has been no large study of the incidence or prevalence of those diseases in a general population. Table 1 summarizes the characteristics of the reviewed patients with IgA-IRGN and SAGN. According to the studies which included patients who had undergone a renal biopsy, of those patients, the frequency of IgA-IRGN was 0.40%, and the frequency of SAGN was 0.45%. In patients with IgA deposition confirmed by a renal biopsy, 4.31% of the patients were diagnosed with IgA-IRGN and 1.66% were diagnosed with SAGN. Among the patients diagnosed with IRGN, 10.18% of the cases were IgA-IRGN and 11.90% were SAGN.

Although a predominant age of patients with either IgA-IRGN or SAGN has not been established, the mean age of the reported IgA-IRGN patients was 54.7 years (range 3–90 years) and that of the SAGN patients was 57.4 (range 6–90 years). IgA-IRGN and SAGN have both been more common in men; 74.5% of the IgA-IRGN patients and 78.4% of the SAGN patients have been male.

Although all of the patients in the first reported study of IgA-IRGN had diabetes mellitus as an underlying disease [108], our review indicates that 42.1% of IgA-IRGN patients and 33.1% of SAGN patients have diabetes mellitus. Other than diabetes, malignant tumors have been one of the causes of infection, but malignant tumors were associated with only 6.2% of the IgA-IRGN cases and 16.4% of the SAGN cases.

### 3.2. Epidemiology and Characteristics

All causative bacteria in the SAGN cases were a staphylococcal strain, and *Staphylococcus aureus* was most frequently detected at 81.7%. In the patients with IgA-IRGN, a staphylococcal strain was most frequently detected (58.4%), but other bacteria or viruses were detected in 26.5% of those patients.

The infectious features of the patients with IgA-IRGN or SAGN are listed in Table 2. Regarding the infection sites, various types of infection have been described: skin infections, cellulitis or superficial abscesses, endocarditis, osteomyelitis or joint infection, upper respiratory infection or pneumonia, deep visceral abscesses, and others.

The IgA-IRGN and SAGN patients’ infections were typically still ongoing when the patients were encountered; however, the length of time from the infection to the onset of the disease varied, with a mean of 24.9 days (range 0–140 days) in the patients with IgA-IRGN and 28.2 days (range 2–140 days) in the patients with SAGN.

### 3.3. Clinical Renal Features

Patients with IgA-IRGN or SAGN usually present with rapidly progressive glomerulonephritis (RPGN) or acute kidney injury (AKI)—79.1% in IgA-IRGN and 75.0% in SAGN (Table 3). Various degrees of proteinuria and microscopic hematuria were present in most cases, and nephrotic-range proteinuria was often present in 59.1% of the IgA-IRGN patients and 52.4% of the SAGN patients. Several patients with IgA-IRGN (40.7%) or SAGN (29.5%) had a purpuric lower extremity rash that was similar to IgA vasculitis (Henoch–Schönlein purpura); a skin biopsy revealed leukocytoclastic vasculitis.

### 3.4. Laboratory Findings

As an infectious disease, SAGN has shown elevations in the white blood cell counts, erythrocyte sedimentation rates, and serum C-reactive protein levels (Table 3). Various degrees of proteinuria and hematuria and decreased renal function (elevated levels of serum creatinine, etc.) were also observed in SAGN—as seen in RPGN and acute kidney injury (AKI). The mean level of serum creatinine in the reports of IgA-IRGN was 3.54 mg/dL (range 0.38–21.94) and that in SAGN was 3.61 mg/dL (range 0.38–10.4). The mean excretion of urinary protein was 4.79 g/day (range 0–19.06) in IgA-IRGN and 4.68 g/day (range 0–16.0) in SAGN.

A polyclonal elevation of serum IgA was often observed in patients with IgA-IRGN (76.0%) or SAGN (78.0%); the mean serum IgA level was 642.5 mg/dL (range 97–1850) mg/dL in IgA-IRGN, and that in SAGN was 685.4 mg/dL (range 97–1850 mg/dL). The serum complement levels observed in IgA-IRGN or SAGN have varied. Decreased serum C_3_ levels were observed in 33.3% of patients with IgA-IRGN and in 34.3% of patients with SAGN. Although rheumatoid factor, autoantibodies (i.e., antiglomerular basement membrane antibody, anti-DNA antibody, and antinuclear antibody), and cryoglobulin were not usually detected in the IgA-IRGN or SAGN cases, antineutrophil cytoplasmic antibody (ANCA) was detected in 5.1% of patients with IgA-IRGN and in 11.4% of patients with SAGN, especially in patients with bacterial endocarditis [8,12].

## 4. Histological Findings

### 4.1. Light Microscopy Findings

Light microscopy examinations have revealed a variety of types of mesangial and/or endocapillary proliferative glomerulonephritis with various degrees of crescent formation and tubulointerstitial nephritis in SAGN (Figure 1a–c). Endocapillary proliferation was more dominant in patients with IgA-IRGN (71.7%) compared to those with SAGN (58.6%), but the rates of mesangial hypercellularity/proliferation were very similar (73.9% in IgA-IRGN; 64.1% in SAGN). Necrotizing and crescentic glomerulonephritis was present in ~10% of cases of IRGN (9.1% in IgA-IRGN; 10.7% in SAGN), and crescent formation was demonstrated in ~50% of both types of IRGN (56.1% in IgA-IRGN; 47.6% in SAGN).

### 4.2. Immunofluorescence Findings

Immunofluorescence has revealed IgA, IgG, and C_3_ in immune complex deposits, typically in the mesangium and along the glomerular capillary walls (Figure 1d–f). Positive IgA staining was observed in 100% of the reported patients with IgA-IRGN, but only in 85.1% of patients with SAGN. IgG staining was positive in 44.4% of patients with IgA-IRGN and in 51.3% of patients with SAGN. Positive C_3_ staining was documented in 97% of patients with IgA-IRGN and in 90.4% of patients with SAGN. C_3_ staining was typically concurrent with IgA staining, and it was sometimes stronger than the staining for IgA.

### 4.3. Electron Microscopy Findings

Electron microscopy examinations have frequently revealed electron-dense deposits (EDDs) in the mesangial area, but subendothelial and small subepithelial deposits can also occur (Figure 1g–h). As shown by the data in Table 4, EDDs are observed in the mesangial area in 78.2% of the IgA-IRGN cases and in 85.3% of the SAGN cases. Subendothelial and subepithelial deposits can also occur. In the reported patients with IgA-IRGN, the frequency of subendothelial EDDs was 43.1% and that of subepithelial EDDs was 54.3%. In contrast, 47.1% of the patients with SAGN exhibited subendothelial EDDs, and 40.5% of them had subepithelial EDDs. Unlike the observations of poststreptococcal acute glomerulonephritis (PSAGN), large subepithelial deposits (humps) were fewer but were identified in 37.5% of patients with IgA-IRGN and 31.7% of patients with SAGN.

## 5. Pathogenesis

### 5.1. Bacterial Superantigens

Some of the enterotoxins (exotoxins) produced by several bacteria include antigens that have the ability to activate a huge number of T cells in a short amount of time; these antigens are called ‘superantigens’ [112]. In a specific antigen’s recognition, the specific antigen that has been processed by antigen-presenting cells (APCs) binds to the groove of the major histocompatibility complex (MHC) molecules, whereas a superantigen binds directly to the outer part of the MHC molecules without being processed [112]. In contrast, a T-cell receptor (TCR) is composed of variable (V), diversity (D), joining (J), and constant (C) domains, and the VDJ gene segment is referred to as the complementarity determining region 3 (CDR3), which recognizes specific antigens [113]. Upon the recognition of a superantigen by T cells, the T cells bind to this MHC molecule/superantigen complex with specific variable regions of the T-cell receptor β-chain (TCR-Vβ) element for each superantigen via an outer portion that differs from the complementarity-determining region (CDR) [112]. The TCR-Vβ repertoire includes ~26 types, and there are multiple TCR-Vβ regions that correspond to a single superantigen—even if the TCR-Vβ region is specific. Only ~0.0001% of the T cells are thus activated in an adaptive immune response, but superantigen exposure can activate up to 30% of the T cells [114].

Superantigens stimulate resting T cells to proliferate, causing a massive activation of T cells and a subsequent release of T-cell-derived lymphokines (e.g., interleukin [IL]-1, -2, or -6) and cytokines (e.g., tumor necrosis factor-alpha [TNF-α] or interferon-gamma [INF-γ]) [115]. There have been reports of a role of staphylococcal enterotoxin in the pathogenesis of toxic shock syndrome [115] and other autoimmune diseases, including rheumatoid arthritis, Kawasaki disease, Sjögren syndrome, and multiple sclerosis [115]. In these diseases, an increase in the usages of some specific TCR-Vβ regions is thought to be a marker of superantigen-related disease [115]. Twenty-six different staphylococcal derivatives have been established as superantigens: toxic shock syndrome toxin-1 (TSST-1), 11 staphylococcal enterotoxins (SE), and 14 staphylococcal enterotoxin-like proteins [116,117].

In the culture supernatants of blood, urine, sputum, and various fluid samples obtained from our patients with SAGN with dominant IgA deposition, staphylococcal enterotoxin C (SEC) was most frequently detected (in 20 of 23 patients). SEA was detected in eight patients, SEB in three, SED in one, and TSST-1 in 13 patients [3,9,10,11,12,13]. SEC and TSST-1 were also detected in SAGN patients with dominant IgA deposition in case reports other than ours [47,58,71]. In analyses of 17 types of TCR-Vβ regions of peripheral blood mononuclear cells (PBMCs) that were obtained from SAGN patients with dominant IgA deposition (Figure 2), the percentages of Vβ8- and Vβ12.1-positive cells were also significantly higher than those observed in healthy controls [3,9,10,11,12,13].

Specific TCR-Vβ usage was also detected in SAGN cases in a report other than ours [47]. The results of a serum cytokine analysis (Figure 3) demonstrated significantly higher serum levels of IL-2, IL-6, IL-8, IL-10, and TNF-α in SAGN patients with dominant IgA deposition compared to those of healthy controls [9,10,11,12,13]. Cytokinemia was also detected in SAGN cases in case reports other than ours [58]. By enzyme-linked immunosorbent assays using anti-C3d antibodies and either antihuman IgG or IgA antibodies, the amounts of both the circulating immune complexes containing IgG and those containing IgA were significantly increased in our patients of SAGN with dominant IgA deposition compared to those in the healthy controls [3].

It has thus been speculated that staphylococcal enterotoxins activate T cells as a superantigen via specific TCR-Vβ regions, and the subsequent excessive release of cytokines activates not only T cells but also B cells; the polyclonal immunoglobulin production leads to the formation of immune complexes, resulting in the onset of SAGN with dominant IgA deposition (Figure 4). In contrast, the reported cases of IgA-IRGN developed due to infections that were caused by bacteria or viruses for which superantigens had not been confirmed, and it may thus not be possible to say that superantigens are involved in all IgA-IRGN cases. However, IgA-IRGN was recently confirmed in a patient with a severe acute respiratory syndrome coronavirus 2 (SARS-CoV-2) infection [17], and SARS-CoV-2 was recently reported to cause many of the biological and clinical consequences of a superantigen [118].

Specific TCR-Vβ usage in an IgA-IRGN patient with a *Chlamydia pneumonia* infection was also detected [62]. Therefore, in IgA-IRGN associated with bacteria and/or viruses that were not produced by a superantigen, the bacterium and/or virus may act as a superantigenic or superantigen-like pathogen, resulting in cytokinemia and the production of polyclonal immunoglobulin.

It was recently demonstrated in an animal model that the specific TCR-Vβ usage in *S. aureus* infection was the result of using the specific VDJ gene segment [119]. That report suggested that the specific TCR-Vβ usage in an *S. aureus* infection may be recognized by the specific antigen originating from *S. aureus*, instead of from superantigens. In our studies, however, we observed specific TCR-Vβ usage in SAGN patients with dominant IgA deposition, unlike in patients with an *S. aureus* infection without glomerulonephritis [3,9,10,11,12,13]. In addition to the *S. aureus* infection that causes the specific TCR-Vβ usage, more specific TCR-Vβ usage was found in SAGN, suggesting that the onset of SAGN has factors other than the antigen associated with the infection. Further analyses of the uses of the specific VDJ gene segment in SAGN are needed.

It has also been demonstrated that several single-nucleotide polymorphisms (SNPs) on the human leukocyte antigen (HLA) class II region of chromosome six were enriched in the infection group of a genomewide association study of biological specimens from culture-confirmed cases and matched controls [120]. Furthermore, it was reported that some bacterial superantigens had a high affinity with specific HLA class II alleles [121]. We demonstrated a specific usage of TCR-Vβ in SAGN, but MHC molecules (HLA class II), which bind to superantigens on the opposite side of TCR-Vβ, have not been analyzed. It could thus be informative to examine MHC molecules in SAGN.

ANCA has been detected in several patients with IgA-IRGN and/or SAGN, and several research groups have reported expansions of various T cell subsets in patients with ANCA-associated vasculitis [122]. However, other studies (including ours) demonstrated that there was no difference in TCR-Vβ usage in peripheral blood and bronchoalveolar lavage in ANCA-associated vasculitis [123,124]. Expansions of peripheral blood T cell subsets expressing T-cell receptors with ANCA-associated vasculitis are thus a controversial topic. We speculate that ANCA-associated vasculitis may be different from SAGN with dominant IgA deposition, which is associated with circulating staphylococcal superantigens.

### 5.2. Neutrophil Extracellular Traps (NETs)

Regarding one of the pathogeneses of ANCA-associated vasculitis, neutrophil extracellular traps (NETs) have been described [125]. NETs are structures of chromatin fibers that are released by dying neutrophils, which trap and kill extracellular invading microbes [126]. However, this DNA web (which contains a number of antimicrobial peptides including myeloperoxidase and proteinase-3) can also stick to and damage the endothelium of small blood vessels, causing vasculitis [125,127]. NETs may be associated with pathology in several kidney diseases, such as AKI, lupus nephritis, and antiglomerular basement membrane disease [128], but the involvement of NETs in infection-related glomerulonephritis is not yet clear.

### 5.3. Dominant IgA Deposition

The mechanism by which IgA is deposited on the glomerulus, which is a characteristic of IgA-IRGN/SAGN, is not clear. IgA-IRGN/SAGN and IgA nephropathy are similar in that IgA is deposited on the glomerulus, but IgA deposition has not been observed on sclerotic glomeruli in SAGN (which is not similar to IgA nephropathy) [129]. In a proteome analysis of kidney biopsy tissues of SAGN patients reported in 2020, significantly higher levels of monocyte/macrophage proteins, a lower abundance of metabolic pathway proteins, and higher levels of extracellular matrix proteins in SAGN were demonstrated compared to the corresponding values in IgA nephropathy [130].

In patients with IgA nephropathy, galactose-deficient IgA1 (Gd-IgA1) depositions were observed in the glomeruli [131]. Gd-IgA1 depositions were also identified in the glomeruli of patients with IgA-IRGN associated with parvovirus B19 infection [24] and of patients with an infection of unknown origin [18]. However, in a cohort analysis of 12 patients with IgA-IRGN [80], the Gd-IgA1 staining showed very weakly positive or negative findings, unlike IgA nephropathy. It was thus speculated that IgA-IRGN and IgA nephropathy are similar regarding at least some points, such as the deposition of glomerular IgA, but these diseases appear to remain separate disease entities.

## 6. Treatments and Outcomes

### 6.1. Treatments

Because persistent infection is an important component in the onset and progression of both IgA-IRGN and SAGN, the treatments for these diseases are essentially treatments for the infecting bacterium or virus. In particular, staphylococcal infections in SAGN are often intractable and present difficulties in diagnosis, with drug-resistant bacteria, deep infection sites, and weakness of immunity. Achieving the appropriate diagnosis of the infectious disease is thus the initial requirement. The diagnoses of infections based on the results of laboratory tests and blood cultures (and the culturing of microbial isolates in particular) are essential to both the determination of the antimicrobial susceptibility and the identification of effective treatments to control the infection. Modalities including X-rays, CT scans, MRIs, transthoracic ultrasonography, and, if necessary, an esophageal echo examination, are thus helpful for identifying the site of infection.

In cases of *S. aureus* infections, resistance to methicillin is particularly relevant to the selection of antibiotics. For patients with a MRSA infection, polypeptide antibiotics (vancomycin, teicoplanin, etc.) and aminoglycoside antibiotics (arbekacin, etc.) are used at doses that are based on the patient’s renal function. It is necessary to adjust the dose(s) while monitoring the drug concentration(s) in order to prevent side effects. In cases with deep infection sites, a prolonged antibiotics course of up to six weeks might be required, and the removal of infected lesions by surgery or drainage may also be useful.

A well-controlled postinfection treatment with one or more corticosteroids and/or immunosuppressants can result in a rapid improvement of urinary parameters and the retention of renal function. In a cohort of elderly patients, renal lesions improved in only 14% of the patients treated with corticosteroids, and 18% of the patients died due to recurrent sepsis [98]. In another investigation of patients with endocarditis, the mortality of the patients treated with corticosteroids (23.5%) was higher than that of the patients treated with antibiotics alone (10%) [90]. In the 62 case reports of IgA-IRGN or SAGN that we reviewed [14,15,16,17,18,19,20,21,22,23,24,25,26,27,28,29,30,31,32,33,34,35,36,37,38,39,40,41,42,43,44,45,46,47,48,49,50,51,52,53,54,55,56,57,58,59,60,61,62,63,64,65,66,67,68,69,70,71,72,73,74,75], 34 of 66 patients were treated with corticosteroids. Among the 34 patients treated with corticosteroids, four patients (12%) died, ten patients (29%) developed end-stage kidney disease (ESKD), and 14 patients (41%) achieved remission. In contrast, among the 32 patients treated without corticosteroids, two patients (6%) died, four patients (13%) developed ESKD, and 14 patients (44%) achieved remission. It thus appears that corticosteroids and/or immunosuppressants usually exacerbate the infection and may cause its recurrence.

Alternative treatments for IgA-IRGN or SAGN have been reported; the efficacy of plasma exchange therapy for removing the immune complexes involved in the development of SAGN [71] and the utility of endotoxin adsorption therapy with polymyxin-immobilized fiber [109] have been described.

### 6.2. Outcomes

The prognoses of IgA-IRGN and SAGN have not yet been established, but the outcomes may be dependent on the eradication of the infection. Our review found that among patients with IgA-IRGN, 17.1% (34/199) died from the disease and 27.9% (50/179) suffered end-stage renal failure. Similarly, 17.6% of patients with SAGN (16/91) died and 31.0% (22/71) suffered end-stage renal failure. Diagnosis and treatment with the appropriate antibiotics are often delayed in cases with deep infection sites, which allows chronic damage to accrue. The risk factors for poor renal outcomes may be similar to those of infectious disease, i.e., advanced age, diabetes mellitus, cancer/malignant tumor, and deteriorated renal function before presentation.

## 7. Conclusions

IgA-IRGN and SAGN are considered to be similar diseases in that most cases of IgA-IRGN are due to infection with *S. aureus*, and most cases of SAGN exhibit IgA deposition in glomeruli. Although positive IgA and C_3_ staining on immunofluorescence is an essential finding in IgA-IRGN, weak or no IgA staining has been observed in 25% of SAGN cases. In contrast, *S. aureus* infection is an essential finding in SAGN, but IgA-IRGN can occur with infections other than *S. aureus*. In analyses of the pathogenic mechanism of both diseases, researchers should keep in mind the commonalities and differences between IgA-IRGN and SAGN.

## Figures and Tables

**Figure 1 ijms-23-07482-f001:**
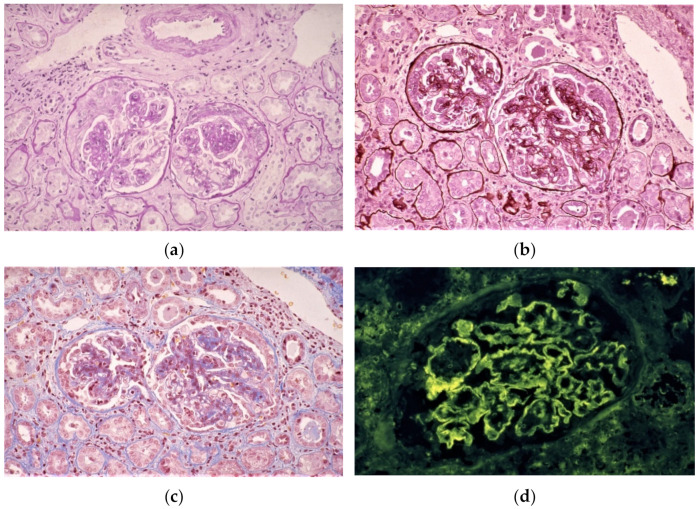
Histological findings of SAGN with IgA-dominant deposition. Light microscopy images of renal biopsy samples stained with periodic acid Schiff (**a**), periodic acid methenamine silver (**b**), and Masson’s Trichrome (**c**); 400× magnification. The images (**a**–**c**) show mesangial and endocapillary proliferative glomerulonephritis with crescent formation and tubulointerstitial nephritis. (**d**–**f**) Immunofluorescence images of renal biopsy samples stained with anti-IgG (**d**), anti-IgA (**e**), and anti-C_3_ (**f**) antibodies, showing IgG, IgA, and C_3_ deposition in the mesangium and along the glomerular capillary walls. Electron microscopy images of renal biopsy samples (**g**,**h**) show electron-dense deposits in the mesangial and subendothelial areas.

**Figure 2 ijms-23-07482-f002:**
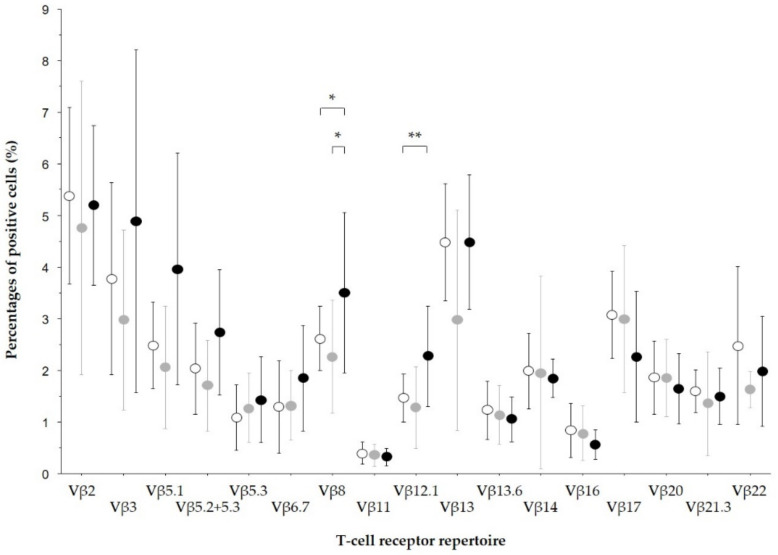
TCR-Vβ usages in SAGN with IgA-dominant deposition. Peripheral blood mononuclear cells (PBMCs) obtained from patients with SAGN, S. aureus-infected patients without glomerulonephritis, and healthy controls stained with fluorescence-labeled monoclonal antibodies against 17 types of TCR-Vβ (6 of 17 types of TCR-Vβ were reported [3,9,10,11,12,13], and the other 11 types are unpublished data); Vβ5.1 (Vβ5c), Vβ5.2+5.3 (Vβ5a), Vβ 5.3 (Vβ5b), Vβ6.7 (Vβ6a), Vβ8 subfamily (Vβ8a) and Vβ12.1 (Vβ12a), Vβ2, Vβ3 (Vβ3a), Vβ11, Vβ13 (Vβ13a), Vβ13.6, Vβ14, Vβ16, Vβ17, Vβ20, Vβ21.3, and Vβ22. The stained PBMCs were then analyzed using flow cytometry. The percentages of 17 TCR-Vβ-positive cells among circulating CD3-positive cells are shown. *Black closed circles*: the mean percentages of cells in patients with SAGN. *Gray closed circles*: the mean percentages of cells in S. aureus-infected patients without glomerulonephritis. *Open circles*: the mean percentages of cells in healthy controls. *Bars*: the standard deviation of positive cells. The percentages of Vβ8- and Vβ12.1-positive cells in the SAGN patients were significantly increased compared to those in the controls, and the percentages of Vβ8-positive cells in the patients with SAGN were significantly higher compared to those in patients without glomerulonephritis (* *p* < 0.05; ** *p* < 0.005).

**Figure 3 ijms-23-07482-f003:**
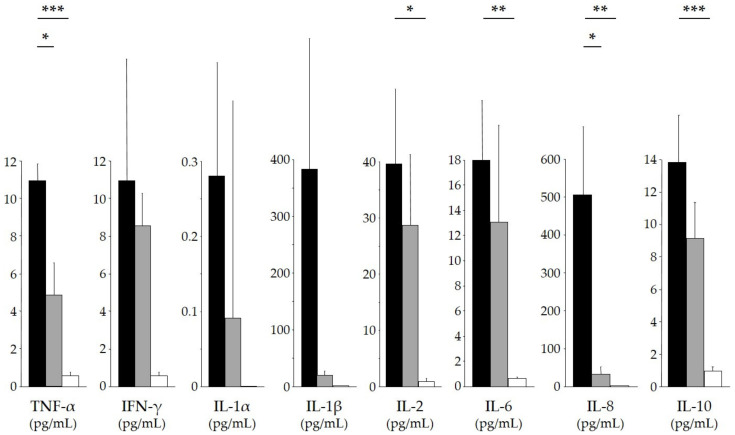
Serum cytokine levels in SAGN with IgA-dominant deposition. Sera were obtained from patients with SAGN *S. aureus*-infected patients without glomerulonephritis and healthy controls, and the following 10 cytokines were measured by enzyme-linked immunosorbent assays: IL-1α, -1β, IL-2, IL-4, IL-6, IL-8, IL-10, TNF-α, TNF-β, and INF-γ [9,10,11,12,13]. The serum levels of eight cytokines are shown. *Black closed bars*: the mean levels of cytokines in patients with SAGN. *Gray closed bars*: the mean levels of cytokines in *S. aureus*-infected patients without glomerulonephritis. *Open bars*: the mean levels of cytokines in healthy controls. *Bars*: the standard deviation. Serum levels of IL-2, IL-6, IL-8, IL-10, and TNF-α in the patients with SAGN were significantly higher compared to those in the controls, and serum levels of IL-8 and TNF-α in the patients with SAGN were significantly higher compared to those in patients without glomerulonephritis (* *p* < 0.05; ** *p* < 0.01; *** *p* < 0.001). Serum IL-4 and TNF-β were not detected in all three groups.

**Figure 4 ijms-23-07482-f004:**
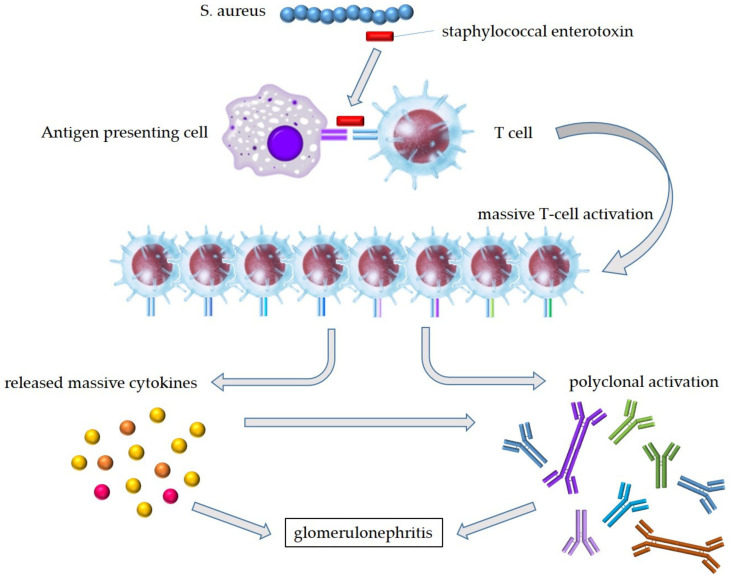
Hypothesis of pathogenesis in SAGN with IgA-dominant deposition. Staphylococcal enterotoxins produced by *S. aureus* bind to specific TCR-Vβ on T cells and the outer part of the MHC molecules without being processed. Bacterial superantigens lead the MHC-unrestricted huge T-cell activation. The subsequent excessive release of cytokines activates not only T cells but also B cells; the polyclonal immunoglobulin production leads to the formation of immune complexes, resulting in the onset of SAGN. *S. aureus*—*Staphylococcus aureus*.

**Table 1 ijms-23-07482-t001:** Characteristics of the reported patients with IgA-IRGN or SAGN.

	IgA-IRGN *n* = 336		SAGN *n* = 218	
Incidence:				
In patients with biopsy	0.40%	117/29,562	0.45%	96/21,257
In patients with IgA deposition	4.31%	40/927	1.66%	12/722
In patients with IRGN	10.18%	34/334	11.90%	5/42
Mean age, years	54.7	3–90	57.4	6–90
Male gender	74.8%	237/317	78.4%	171/218
Underlying disease:				
Diabetes	42.1%	118/280	33.1%	52/157
Cancer	6.2%	12/194	16.4%	11/67

IRGN—infection-related glomerulonephritis; SAGN—Staphylococcus infection-associated glomerulonephritis.

**Table 2 ijms-23-07482-t002:** Infectious features of the reported patients with IgA-IRGN or SAGN.

	IgA-IRGN *n* = 336		SAGN *n* = 218	
Causative bacteria				
Staphylococcal strain:	58.4%	185/317	100.0%	218/218
*S. aureus*	54.9%	174/305	81.7%	178/218
*S. epidermidis*	1.7%	5/298	6.9%	15/218
Other Staphylococcus	1.7%	5/298	11.5%	25/218
Streptococcus strain	5.5%	17/310	0.5%	1/211
Other bacteria	21.0%	65/310	3.3%	7/211
Unknown/not detected	23.6%	73/310	0%	0/211
Infection site:				
Cellulitis/skin infection	25.1%	68/271	23.6%	45/191
Endocarditis	4.1%	11/271	12.6%	24/191
Osteomyelitis/joint infection	11.4%	31/271	17.3%	33/191
Respiratory infection	18.8%	51/271	12.0%	23/191
Visceral abscess	9.2%	25/271	15.7%	30/191
Others	22.9%	62/271	19.4%	37/191
Unknown	17.0%	46/271	0%	0/271

**Table 3 ijms-23-07482-t003:** Clinical features and laboratory data of the reported patients with IgA-IRGN or SAGN.

	IgA-IRGN *n* = 336		SAGN *n* = 218	
Clinical features:				
AKI or RPGN	79.1%	178/225	75.0%	78/104
Nephrotic syndrome	59.1%	101/171	52.4%	75/143
Proteinuria	98.9%	183/185	98.4%	183/186
Hematuria	95.0%	226/238	91.4%	106/116
Purpura	40.7%	46/113	29.5%	49/166
Laboratory data:				
Excretion of urinary protein, g/day	4.79 (*n* = 272)	0–19.06	4.68 (*n* = 125)	0–16.0
Serum creatinine level, mg/dL	3.54 (*n* = 312)	0.38–21.94	3.61 (*n* = 138)	0.38–10.4
Serum IgA level, mg/dL	642.5 (*n* = 98)	97–1850	685.4 (*n* = 77)	97–1850
Elevated serum IgA level	76.0%	73/96	78.0%	46/59
Decreased serum C_3_ level	33.3%	74/222	34.3%	60/175
Decreased serum C_4_ level	11.0%	18/163	15.5%	26/168
Positive test for ANCA	5.1%	5/99	11.4%	12/105

AKI—acute kidney injury; ANCA—antineutrophil cytoplasmic antibody; RPGN—rapidly progressive glomerulonephritis.

**Table 4 ijms-23-07482-t004:** Histological findings of the reported patients with IgA-IRGN or SAGN.

	IgA-IRGN *n* = 336		SAGN *n* = 218	
Light microscopy:				
Mesangial hypercellularity	73.9%	209/283	64.1%	66/103
Endocapillary proliferation	71.7%	203/283	58.6%	106/181
Membranoproliferative GN	5.9%	16/270	7.8%	8/103
Necrotizing/crescentic GN	9.1%	21/231	10.7%	11/103
Presence of crescents	56.1%	143/255	47.6%	78/164
Immunofluorescence:				
Positive staining with IgG	44.4%	118/266	51.3%	79/154
Positive staining with IgA	100.0%	303/303	85.1%	149/175
Positive staining with C_3_	97.0%	291/300	90.4%	161/178
Electron microscopy:				
Subepithelial EDD	54.3%	119/219	40.5%	62/153
Subendothelial EDD	43.1%	81/188	47.1%	33/70
Intramembranous EDD	15.6%	31/199	6.9%	4/58
Mesangial EDD	78.2%	147/188	85.3%	64/75
Humps	37.5%	90/240	31.7%	46/145

EDD—electron dense deposit; GN—glomerulonephritis.

## Data Availability

Not available.

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
