# Peer review of "Staphylococcus aureus Infection-Related Glomerulonephritis with Dominant IgA Deposition"

_ijms, 2022, doi:10.3390/ijms23137482_

Round 1

Reviewer 1 Report

Dear Authors,

Thanks for this manuscript, what fits to improve the knowledge into the specific field. Especially nice is described M+M part 2.1 what seldomly is so nicely developed.

I have just some few objections;

1) Fig.1. requests controls, - could you add them, please?

2) please, add some EM micrographs, too for the subsection 4.3., if possible; I do not insist, but they would fit here very well;

3) Please, decipher the abbreviations for the Fig.4;

4) please, make shorter and more precise the Conclusions; also remove the personalization from this part;

5) well, there are some 17 references from the previous century, - could you exchange them or remove, please; the other option is to mention everywhere , where you reference them, the historical aspect...

Author Response

Point 1: Thanks for this manuscript, what fits to improve the knowledge into the specific field. Especially nice is described M+M part 2.1 what seldomly is so nicely developed. I have just some few objections; 1) Fig.1. requests controls, - could you add them, please?.

Response 1: Thank you for your thoughtful comments and recommendations for our manuscript. S. aureus-infected controls were defined as the patients with S. aureus infection but no urinary protein and urinary occult blood using dip-and-read sticks (Multistix®: Sankyo Co. Ltd., Tokyo, Japan). Therefore, No histological examination was performed on S. aureus-infected patients without glomerulonephritis.

Point 2: please, add some EM micrographs, too for the subsection 4.3., if possible; I do not insist, but they would fit here very well;

Response 2: As the reviewer recommends, we added pictures of electron microscopy in Fig.1.

Point 3: Please, decipher the abbreviations for the Fig.4;

Response 3: As the reviewer recommends, we additionally describe the full spelling of the abbreviation.

Point 4: please, make shorter and more precise the Conclusions; also remove the personalization from this part;

Response 4: As the reviewer recommends, we rewrote the conclusion part as concisely as possible.

Point 5: well, there are some 17 references from the previous century, - could you exchange them or remove, please; the other option is to mention everywhere, where you reference them, the historical aspect...

Response 5: As the reviewer recommends, we have changed old references that can be replaced to new ones as much as possible. We have also removed two pre-2000 cohort studies that are neither IgA-IRGN nor SAGN and explained that in the method section. However, even in the references before 2000, the ones of IgA-IRGN or SAGN are necessary for this analysis, so they are left as they are.

Reviewer 2 Report

The paper of Mamiko Takayasu and colleagues: “ Staphylococcus aureus infection-related glomerulonephritis with dominant IgA deposition” is an interesting and well-written review article.

The abstract is a good summary of article and invites to read the entire manuscript.

In the Introduction the Authors introduce the topic, present the current state of the research and cite key publiactions.

Review methods are well described.

Clinical features are easy to read, tables are well-organized and helpful.

All presented figures are attractive and are an important part of review article. They hold the interest of readers.

The reference list is well organized and contains current publications related to the topic.

The manuscript is written in a good style and easy to read, there are no grammatical,  punctuation or linguistic errors.

In my opinion the paper is valuable to publish in Int J Mol Sci.

Author Response

Response to Reviewer 2 Comments

Point 1: The paper of Mamiko Takayasu and colleagues: “Staphylococcus aureus infection-related glomerulonephritis with dominant IgA deposition” is an interesting and well-written review article.

The abstract is a good summary of article and invites to read the entire manuscript. In the Introduction the Authors introduce the topic, present the current state of the research and cite key publiactions. Review methods are well described. Clinical features are easy to read, tables are well-organized and helpful. All presented figures are attractive and are an important part of review article. They hold the interest of readers. The reference list is well organized and contains current publications related to the topic. The manuscript is written in a good style and easy to read, there are no grammatical, punctuation or linguistic errors. In my opinion the paper is valuable to publish in Int J Mol Sci.

Response 1: Thank you for your thoughtful and kind comments for our manuscript..

This manuscript is a resubmission of an earlier submission. The following is a list of the peer review reports and author responses from that submission.

Round 1

Reviewer 1 Report

Dear Authors,

Despite weakness on design and structure of the text, I found that this article can provide a significant useful overview, but just in case of its serious improvement:

1) The structure is not enough clearly divided, with data not explained enough or overlapping among sections. So, Introduction misses the definition, incidence and significance, also the realistic research status in the scientific literature; No 3 is not pathological, but Morphopathological findings; No 4 is extra or should come after the No 3.

2) According to the manuscripts, it is not clear the review method described. I would suggest include a “Method” section, describing how the comprehensive search of publications has been performed (eg: database used, syntax applied, key words, period of the search …etc.). I would also invite the authors to describe the number of citations found, as well as how the selection process of relevant papers included was done.

3) Decipher all the abbreviations, please at the Figs, tables;

4) Conclusions are not valid as actually they represent the part of main text. please, move them to the review main part and develop short, precise Conclusions;

5) 25 references for the review paper (from which 9 belongs to the previous century) is catastrophically small number and not for the review manuscript! Please, expand the parts requested and re-place the old one references to new one!

Reviewer 2 Report

Dear Authors!

Congratulations on your study and your interest in such a rare condition, which in fact may be not sa rare as it is believed. You approached the topic in detail, of course, there might be some controversies or data lack that you mention (e.g. genetic differences between the nations or lack of MHC analysis), however the study and the data need to be published, and those are only minor shortages. Besides, the discussion underlines them and it shoun not be treated as a drawback- at this stage we cannot analyse it all. 

If I make a small suggestion, an international call-for-data or (even better) a creation of an international working-group could distinctly accelerate the studies and expand our knowledge on this topic. 

There are some minor changes I would recommend:

  • first, state clearly the topic of the paper (is the paper prepared to review the subject or to show your study data, etc. )
  • if it is possible, I realize that a detailed formally-organized systematic review requires slightly different approach to the paper, however, it increases the number of citations, and in this case you actually DID a systematic review- it would not require much more work, but would add value if the tile included systematic review. This is up to you. 

Best regards

Reviewer 3 Report

Takayasu et al wrote a nice overview on Staphylococcus associated glomerulonephritis. The review part of the manuscript is very well done and I congratulate the authors with this. I have no significant additions to that part. The fact that the authors present new data comes a bit surprising. I think to make the manuscript publishable either the data could just be referred to without summing up all the antibodies used with the correct citation. Alternatively the authors could leave it this way but include a short "methods" section. Either way I think the manuscript is well written and publishable.

Reviewer 4 Report

The manuscript entitled “Staphylococcus infection-associated glomerulonephritis: a role of staphylococcal enterotoxin in the pathogenesis” is a review article using 3 previous reports and 3 their own papers to provide a hypothesis for the pathogenesis of SAGN. Although their studies have been done well and published, the clinical implication and novelty of the review have not been shown. Besides, there are some problems about the figures and table in the manuscript needed to be solved before resubmitting.

Major revisions:

  1. This manuscript looks more like a research article but not review article. The organization is alright but discussion in each section is not well written. It focuses on 7 studies of which include their own 3 studies. However, the authors’ results were mostly different from those of the other previous studies. But they did not discuss the reason why. Besides, some of the other studies had been cited and discussed in one pool analysis of published literature (Wang, SY, et al. Medicine 95(15):e3386, 2016) and one review article (Nat Rev Nephrol 16:32–50, 2020). Both could be used as templates for good presentation of the similar study.
  2. Some the sections discuss their own studies but not others recently reported, such as, Front Microbiol 2019, 10:2012. In addition, the authors transform their previous results (Nephrol Dial Transplant 2000 15: 1170–1174) into two figures (figure 2 & 3).
  3. Table should be reorganized. The numbers are very odds and not equal to total numbers. The results of bacteremia are dramatically different between previous studies and the authors’. Is it possible that the definition is totally different? May meta-analysis be used to rewrite the article. The description in the text and data in table 1 are not the same, such as “The serum complement levels in SAGN were within normal limits in “all” of our patients”. But 4 of them had “decreased C3 levels” in Table 1.
  4. The quality of Figure 1 is too bad to be re-drawn. The labels should be added according to figure legend.
  5. Why did the authors compare the T-cell receptor and serum cytokine analysis between SAGN and normal control only but no other causes of GN? These levels would be elevated in any infectious diseases.

Minor revision:

  1. English editing is needed.
